# The Development of a Digital Twin to Improve the Quality and Safety Issues of Cambodian Pâté: The Application of 915 MHz Microwave Cooking

**DOI:** 10.3390/foods12061187

**Published:** 2023-03-11

**Authors:** Sovannmony Nget, Hasika Mith, Géraldine Boué, Sébastien Curet, Lionel Boillereaux

**Affiliations:** 1Oniris, Nantes Université, CNRS, GEPEA, UMR 6144, F-44000 Nantes, France; 2RIC, Institute of Technology of Cambodia, Russian Federation Blvd., Phnom Penh P.O. Box 86, Cambodia; 3Oniris, INRAE, SECALIM, F-44300 Nantes, France

**Keywords:** microwave, heat transfer, mass transfer, Lewis analogy, Cambodian pâté, 915 MHz

## Abstract

Foodborne diseases are common in Cambodia and developing good food hygiene practices is a mandatory goal. Moreover, developing a low-carbon strategy and energy efficiency is also a priority. This study focuses on pâté cooking, a very common food product in Cambodia. In this paper, the authors chose to develop a digital twin dedicated to perfectly predict the temperature for cooking in a 915 MHz single-mode cavity, instead of using a classical and energy-consuming steaming method. The heating strategy is based on a ramp-up heating and a temperature-holding technique (with Tylose^®^ as the model food and Cambodian pâté). The model developed with COMSOL^®^ Multiphysics software can accurately predict both local temperatures and global moisture losses within the pâté sample (*RMSE* values of 2.83 and 0.58, respectively). The moisture losses of Cambodian pâté at the end of the process was 28.5% d.b (dry basis) after a ramp-up heating activity ranging from 4 to 80 °C for 1880 s and a temperature-holding phase at 80 °C for 30 min. Overall, the accurate prediction of local temperatures within Cambodian pâté is mainly dependent on the external heat-transfer coefficient during the temperature-holding phase, and is specifically discussed in this study. A 3D model can be used, at present, as a digital twin to improve the temperature homogeneity of modulated microwave power inputs in the future.

## 1. Introduction

Thermal treatment is an important process for solid food products to fulfill the microbial safety requirements and their organoleptic criteria. Temperature control is key to achieve those criteria. Therefore, the choice of cooking method is the key to ensure the achievement of the required temperature. Pâté is one of the most popular solid foods that can be found in almost every market in Cambodia. Traditionally, the cooking method used in its production is steaming; however, this cooking method is time- and energy-consuming. In the search for an alternative cooking method with a lower carbon footprint, microwave heating has been proposed due to its advantages, such as flexibility and the possibility to achieve high heating rates with high energy efficiency [1]. To fully evaluate the application of this heating method for cooking Cambodian pâté, it is necessary to know the temperature profile of each point inside the product, which can be achieved thanks to the predictive model for the microwave heating of this product. The temperature profile within the sample is important in terms of an microbial inactivation evaluation.

The microwave-heating modeling of food materials is a subject that has been explored by several research investigations [2]. The application of multi-mode microwave cavities in food treatment has been widely explored by many studies, most of which focus on the domestic microwave cavity, such as the studies by Wipawee, Witchuda, and Pitiya; Klinbun and Rattanadecho [3]; Pitchai et al. [4]; Zhu et al. [5]; Shen et al. [6]; Chen et al. [7]; and Gulati, Zhu, and Datta [8]. Further studies on the advanced modeling of the microwave-heating process for domestic applications were published by combining electromagnetism with heat and mass transfer in porous materials, which was applied for rotating grain-drying simulations and the microwave heating of frozen, mashed potatoes using COMSOL^®^ Multiphysic. Interestingly, the study by Shen et al. [6] used the Chilton–Colburn analogy to account for the mass transfer from the convective heat-transfer coefficient at the surface.

For a static multi-mode microwave applicator, such as the household microwave oven, a cavity is characterized by a non-uniform electric-field pattern due to multiple wave reflections between the sample and metallic walls. These could lead to undesirable hot-temperature spots that can be slightly reduced by the use of a mode stirrer or rotating plate [9]. In the case of a single-mode microwave cavity, the shape of the incident electric field is well-known and the materiel under investigation can be placed at the location where a maximum electromagnetic field intensity provides the best thermal processing performance [10].

In terms of microwave power generation, the study by Zhou et al. [11] showed that the microwave frequency provided by a magnetron-based generator has a broader spectrum and depends on the sample’s position and product’s physical properties, while the solid-state microwave generator has a better spectral quality resulting in a stable heating pattern for the processed food.

To seek an alternative treatment for Cambodian food products, this study aims to investigate the feasibility of the application of microwave-heating technology for cooking Cambodian pâté. To improve the microwave penetration depth and provide the best heating performances, the originality of this study lies in its use of a 915 MHz microwave frequency generated by a solid-state microwave generator within a single-mode microwave cavity. The developed model incorporated electromagnetic waves, coupled with heat and mass transfer, to predict the temperature and moisture concentration profiles during the microwave cooking of Cambodian pâté. The distinct design of the waveguide transition (including the antenna), impedance-matching elements, single-mode microwave cavity, and sliding short plunger also contribute to the state-of-the-art nature of this study. A food simulant (Tylose^®^) is used as a preliminary experimental validation of the model due to its more uniform composition. The model is then improved and validated further to account for the mass transfer phenomena during the microwave heating of raw Cambodian pâté. The effect of the external natural convective heat-transfer coefficient is also discussed, based on the temperature profiles of the samples.

## 2. Materials and Methods

### 2.1. Tylose Preparation

A Tylose^®^ sample was used for the validation of the electromagnetic and heat-transfer models due to its homogenous and simple composition, which allowed us to avoid potential uncertainties concerning the heterogeneity of the pâté and to guarantee good reproducibility. The process of producing Tylose^®^ was similar to previous research investigations [12] with some modifications. The sample was created by slowly mixing an 11% mass-to-mass ratio of methyl 2-hydroxyethyl cellulose (Tylose MH 1000 YP2, Shin Etsu, Ibaraki, Japan) with water at 90 °C, while continuously stirring it with a magnetic stirrer with some additional help from a small, glass rod. After the mixture was well mixed, it was poured into a polyethylene terephthalate (PET) microbiology sample container cup with a diameter of 5.2 cm and height of 6.5 cm. The cylindrical shape of the sample was selected to simplify the preparation process and reduced the overheated parts of the sample submitted to microwaves as compared to other shapes, such as cubes, cuboids, and spheres [13]. The sample was stored at 4 °C overnight before it was used in the experiment.

### 2.2. Cambodian Pâté Preparation

The pâté sample was created by using 500 g of pork loin. The tendon of the meat was removed, and the meat was cut into small pieces. The ingredient mixture was created by mixing 5 g of table salt (“Gros sel Marin″ from the Perrosel company, Le Fenouiller, France), 4.1 g of sugar (from the supermarket, France), 10 g of cooking oil (Rustica brand, Paris, France), 1.63 g of black pepper (from the supermarket, France), 1 g of baking soda (from “La Baleine″ company, Paris, France), 17.5 g of corn starch (“Tablier blanc″ brand, Paris, France), 3.34 g of garlic powder (Rustica brand, Paris, France), 9 g of fish sauce (Suzi Wan, France), and 120 g of tap water. The cut meat was put into a blender (Thermomix TM6, Vorwerk, Wuppertal, Germany) and the blending process occurred for 1 min. The ingredient mixture was added to the blended meat and the blending process occurred for 4 to 5 min until the meat was well mixed. The blended mixture was called raw pâté. For the microwave heating, the raw pâté was placed into a polyethylene terephthalate (PET) sample container cup with a wall thickness of 0.1 cm to fill up the final dimension of 5.2 cm in diameter and 6.5 cm in height.

### 2.3. Dielectric Properties’ Measurements

The dielectric properties’ measurements were performed at the frequency of 915 MHz using an open-ended coaxial probe (dielectric kit 85070E, Agilent Technologies, Penang, Malaysia) connected to a vector network analyzer (ENA 5062A, Agilent Technologies, Penang, Malaysia). The calibration method for the open-ended coaxial probe was performed from a one-port calibration procedure by using a mechanical calibration kit where each standard had a precisely known magnitude and phase response as a function of frequency (Open, Short, Load with deionized water). For the dielectric measurements, Tylose^®^ and raw pâté were placed, respectively, inside a small heating chamber where hot water inside an external jacket coming from a water bath (Julabo model F32, Seelbach, Germany) circulates as a closed loop. The coaxial probe was placed in contact with the sample’s surface and the probe was held in this position. The small heating chamber was covered by a thermal insulator and until thermal equilibrium was attained. Since the Tylose^®^ sample started to change its aspect and lost its integrity at 70 °C, the dielectric properties were measured at 915 MHz at temperatures of 4, 10, 20, 30, 40, 50, and 60 °C for the Tylose^®^ sample and an additional 70, 80, and 90 °C for the pâté sample.

### 2.4. Thermal Conductivity Measurements

The thermal conductivity values of the samples (Tylose^®^ and raw pâté) were measured using the transient-line heat-source method (Thermal analyzer TEMPOS with probe KS-3, Meter, München, Germany). The probe was first calibrated by using a known thermal conductivity glycerin (glycerol) solution to obtain reliable measurements with the reference value in the range of the measured values for the Tylose^®^ and pâté. The samples were prepared in a 50 mL centrifugal tube and stored at 4 °C. Since the thermal conductivity of the Tylose^®^ was fairly changed with the changing of the temperature, the thermal conductivity of the Tylose^®^ was measured at the temperature of 20 °C. The thermal conductivity of the pâté was measured by placing a cylindrical container with raw pâté inside the small heating chamber, as described in the dielectric measurement method. The thermal conductivity of the pâté was measured at temperatures of 4, 10, 30, 50, 70, and 90 °C.

### 2.5. Heat Capacity Measurements

The heat capacity was measured experimentally by the differential scanning calorimetry technique with a heat flux calorimeter (µDSC7 Evo, Setaram Instrumentation, Kep Technologies, Mougins, France). The heat capacity of the samples was analyzed from −5 to 90 °C with a scanning heating rate fixed at 1 °C/min. Dedicated software (Calisto, Setaram, Caluire-et-Cuire, France) was used for the temperature program and post-processing of the results. A blank test was first performed with two empty cells (stainless-steel containers). Then, the test was performed with the sample (200 mg introduced to the sample cell) and empty reference cell following the specified temperature program. Two isotherms were applied at both the beginning and end of the temperature program to allow for the specific heat calculation. The measurement was performed in duplicate to obtain an average value.

### 2.6. Microwave Device Configuration

The 915 MHz microwave frequency was generated from a solid-state microwave generator (GLS 1500, SAIREM, Décines-Charpieu, France). A total of 30 W of microwave energy was supplied to the waveguide transition via a 1.5 m-length coaxial cable with 9.7% power loss (obtained from vector network analyzer measurements). Therefore, after passing through the coaxial cable, the effective power supplied to the microwave antenna was 27.1 W.

The dimension of the microwave antenna, iris, sliding short circuit, and detailed features inside the waveguide cavity were measured and the geometry was built in COMSOL^®^ Multiphysics by including all the elements, as shown in Figure 1. The waveguides used were standard WR975 single-mode aluminum waveguides (SAIREM, Décines-Charpieu, France), with a 9.4 cm aperture iris (microwave-coupling device), a single-mode microwave applicator, and a sliding short circuit (SSC), as shown in Figure 1A. The iris was used as a metal plate that contained an opening through which the waves could pass to improve the impedance matching of the microwave cavity. The iris was located in the transverse plane of the electric-field strength to act as an inductive iris where the edges were perpendicular to the magnetic field’s orientation within the waveguide. The bolts and cavity rods were included due to their influence on the electric field inside the cavity [14]. The inner surface of the waveguide, antenna, cavity rod, iris, and bolt in the cavity were all composed of aluminum. However, the antenna feeder and SSC surface properties had the properties of copper and brass, respectively. Moreover, the geometry of the PET plastic container was also included to approximate the experiment. The iris was used in this system due to its ability to focus the electromagnetic wave with negligible energy losses. The electromagnetic wave was transmitted from the antenna through port 1 (TEM mode), while the microwaves were transmitted from the antenna through the waveguide transition (TE10 mode).

### 2.7. Impedance-Matching Characterization

The Tylose^®^ or pâté sample with an initial ambient temperature (20 °C) was placed at positions A, B, C, and D, as shown in Figure 2. The waveguide transition (port 1) was connected to a vector network analyzer with a coaxial cable, and the reflected voltage (then converted to the S11 parameter) was measured for each sample location following the various positions of the sliding short circuit (from 700 to 900 mm from the center of the cavity with a 10 mm increment step). The microwave power-reflection coefficient for each sliding short-circuit position and each sample position was calculated using the following expression:(1)$$RF=\frac{P_{reflected}}{P_{incident}}=10^{\frac{S_{11\left[dB\right]}}{10}}$$

### 2.8. Microwave-Heating Experiments

Before microwave heating, the sample (Tylose^®^ or raw pâté) was kept at 4 °C overnight before it was placed at the center of the single-mode cavity, and the waveguide transition was connected to the microwave generator (GLS 1500, SAIREM^®^, Décines-Charpieu, France). Three optical fibers connected to a data logger (Rugged monitoring H201, Quebec, PQ, Canada) were placed at positions 1, 2, and 3 within the sample (shown in Figure 3A,B) with the help of a polyethylene sensor holder to insure a precise geometrical position (shown in Figure 3C).

The sliding short-circuit position was fixed at 850 and 858 mm for the temperature validation of the pâté and Tylose^®^, respectively, when both samples were placed at position A, and fixed at 563 mm when the Tylose^®^ sample was placed at position B.

Microwave power input values and temperature measurements at the 3 points were recorded during the microwave processing. The microwave power was stopped after the temperature at point 1 reached 60 °C for the Tylose^®^ sample. For the pâté sample, heating consisted of two sequences: a ramp-up heating and temperature-holding phase. During the ramp-up heating phase, 30 W of microwave power was constantly applied until the temperature of point 1 reached 80 °C. Then, the second phase consisted of a microwave power regulation to hold the temperature of point 1 at 80 °C.

### 2.9. Moisture Loss Characterization

Before the experiment, the sample and container were weighed to obtain the initial mass (*m*_0_). The pâté sample was heated up to 80 °C and held at that temperature for 0, 15, and 30 min. After reaching each condition, the microwave power was switched off and the sample was immediately weighed to obtain the final mass (*m_f_*). The recorded microwave power-input profile was used as the input value for the simulation. The moisture loss was calculated as the percentage of dry basis by using Equation (2), where *rd* was the ratio of dry matter in the wet sample (23%):(2)$$\%\;\mathrm{Moisture}\;\mathrm{loss}=\frac{m_{f}-m_{0}}{m_{0}\cdot rd}$$

### 2.10. Infrared Image

The infrared images were obtained by using an infrared camera (Cedip Silver 420M, Muchen, Germany). The camera was mounted on a support system, which was placed on top of the microwave applicator, and the images were captured through the hole of the cavity (port 2 in Figure 1). Before the start of the experiment, the camera was properly calibrated by using a black body. Then, the camera was set to capture the infrared images from the beginning of the process for 5 min for the Tylose^®^ sample and 14 min for the pâté sample. The emissivity of the samples was approximated to 0.9 by measuring the ambient surface temperatures with thermocouples and the emissivity value was adjusted with Altair software (Cedip Infrared Systems, Muchen, Germany).

## 3. Modeling

### 3.1. Assumptions

To develop the electromagnetic model and assess the heat- and mass-transfer phenomena during microwave heating, some assumptions were made:Tylose and raw pâté were considered homogenous and isotropic [10].The samples had homogenous initial temperatures [10].The thermophysical and dielectric properties of Tylose^®^ were constant within the temperature range of the experiment [15].The thermal conductivity and dielectric properties of the pâté were a function of temperature, while the specific heat capacity of the pâté was assumed to be constant within the range of temperatures assessed.Thermal conductivity was constant within the variation range of moisture.Density was considered constant throughout the process [15].The shrinkage of the samples was considered negligible throughout the process [10].The bottom of the sample in contact with the PTFE support was considered thermally insulated.The sample was in perfect contact with the PET container.The ambient air was at a constant temperature (Tair = 293.15 K).The absolute humidity of the air surrounding the sample was constant throughout the experiment due to a small amount of evaporation from the sample into the air inside the microwave cavity.

### 3.2. Modeling of Microwave Propagation

The propagation of microwaves within the dielectric materials with the absence of free charges and currents was governed by Maxwell’s equations [16]:(3)$$\nabla \times H=\frac{\partial D}{\partial t}=\frac{\partial \varepsilon E}{\partial t}$$
(4)$$\nabla \times E=-\frac{\partial B}{\partial t}=-\frac{\partial \mu E}{\partial t}$$
(5)∇·D=∇·εE=0
(6)∇·B=∇·μH=0
where *µ* is the magnetic permeability that is considered as *µ*_0_ = 4*π*· 10^−7^ H/m for the vacuum and *ε* is the complex permittivity with $\varepsilon_{0}=8.854\cdot 10^{-12}\;F/m$ for the vacuum. The relative complex permittivity $\varepsilon_{r}$ is composed of the relative dielectric constant $\varepsilon_{r}^{\prime }$ and the relative dielectric loss factor $\varepsilon_{r}^{\prime \prime }$ with the complex notation “*i*″ as shown in Equation (7):(7)$$\varepsilon =\varepsilon_{0}\varepsilon_{r}=\varepsilon_{0}\left(\varepsilon_{r}^{\prime }-i\varepsilon_{r}^{\prime \prime }\right)$$

The Helmholtz equations are used to describe the propagation of the electric (Equation (8)) and magnetic (Equation (9)) fields inside a dielectric medium, and *ω* is the angular frequency (rad/s):(8)$$\nabla^{2}\vec{E}+\mu_{0}\mu_{r}\omega^{2}\varepsilon_{0}\left(\varepsilon_{r}^{\prime }-i\varepsilon_{r}^{\prime \prime }\right)\;\vec{E}=0$$
(9)$$\nabla^{2}\vec{H}+\mu_{0}\mu_{r}\omega^{2}\varepsilon_{0}\left(\varepsilon_{r}^{\prime }-i\varepsilon_{r}^{\prime \prime }\right)\;\vec{H}=0$$

-Initial condition *E* = 0, *t* = 0.-The boundary condition for the TE10 mode:*Ex* = 0 at *y* = 0;*Ex* = 0 at *y* = 12.38 cm (width of the waveguide);*Ey* = 0 at *x* = 0;*Ey* = 0 at *x* = 24.76 cm (height of the waveguide);*Ez* = 0.-The impedance boundary condition is used at the walls of the waveguides for brass, copper, and aluminum metallic surfaces [17]


(10)
$$\sqrt{\frac{\mu_{0}\mu_{r}}{\varepsilon_{0}\varepsilon_{r}-\frac{i\sigma_{material}}{\omega }}}n\times H+E-\left(n\cdot E\right)n=\left(n\cdot E_{s}\right)n-E_{s}$$


These surfaces were not considered as perfect conductors. Therefore, the electric field could penetrate these materials at a very short distance, which distributed a very small amount of microwave power loss depending on their electrical conductivities.

-Perfect electrical conductor: apply to the surface of the metallic core inside the PTFE ring of the antenna (*n* × *E* = 0).-Port boundary condition: the percentage values of microwave reflected power (*RF*) at the input port (coaxial port for antenna) and port 2 were calculated from the squared magnitude values of *S*_11_ and *S*_21_ [16]. The *S*-parameter values at both ports are:


(11)
$$S_{11}=\frac{\int_{port1}^{\;}\left(\left(E_{c}-E_{1}\right)\cdot E_{1}{}^{*}\right)dA_{1}}{\int_{port1}^{\;}\left(E_{1}\cdot E_{1}{}^{*}\right)dA_{1}}\;$$



(12)
$$S_{21}=\frac{\int_{port2}^{\;}\left(\left(E_{c}-E_{2}\right)\cdot E_{2}{}^{*}\right)dA_{2}}{\int_{port2}^{\;}\left(E_{2}\cdot E_{2}{}^{*}\right)dA_{2}}$$


The volumetric microwave absorbed power (*Q_M_*) due to dielectric losses within the medium is expressed by the following equation [4,10,18]:(13)$$Q_{M}=\pi f\varepsilon_{0}\varepsilon_{r}^{\prime \prime }E^{2}=\frac{1}{2}\omega \varepsilon_{0}\varepsilon_{r}^{\prime \prime }E^{2}$$
(14)$$P_{ab}={∭}^{\;}Q_{M}dV$$
where *f* is the frequency of the microwave propagation and *P_ab_* is the microwave absorbed power.

### 3.3. Heat Transfer

The heat source term *Q_M_* must be included in the heat conservation equation for the heat-transfer study. However, the sample was also exposed to water evaporation from the top surface and to natural convection at both lateral and upper surfaces during the heating process. Taking into account the heat source term due to microwaves, the energy balance equation is expressed as [19,20,21]:(15)$$\rho_{s}C_{p}\frac{\partial T}{\partial t}=\nabla \cdot \left(k\nabla T\right)+Q_{M}$$
where *ρ_s_* is the density of the sample, *C_p_* is the specific heat capacity of the sample, and *k* is the thermal conductivity of the sample.

-Initial condition: *T*_0_ = 4 °C, *t* = 0.-The boundary condition for the top surface of the sample involves both evaporation and natural convection phenomena [22]:


(16)
$$k\frac{\partial T}{\partial n}=hc\left(T_{ext}-T_{s}\right)+Q_{ev}\;\;\;\forall t>0$$


*Q_ev_* is the heat loss due to evaporation, *hc* is the convective heat-transfer coefficient, *T_s_* is the temperature at the surface of the sample, and *T_ext_* is the temperature of the hot-air film close to the surface of the sample by the effect of natural convection.

-The boundary values for the side surface of PET sample cells are:


(17)
$$k\frac{\partial T}{\partial n}=hc\left(T_{ext}-T_{s}\right),\;\;\;\;\;\;\forall t>0$$


The perfect insulation boundary condition is applied to the bottom of the sample cell.
(18)n·q=0,  ∀t≥0

### 3.4. Mass Transfer

At the top surface of the sample, the evaporative heat loss is calculated using the following equation:(19)$$Q_{ev}=D_{m}\lambda \frac{\partial C}{\partial n}$$
(20)$$D_{m}=\frac{K_{m}}{\rho_{s}C_{m}}$$
where *D_m_* is the moisture diffusivity (m^2^/s) [23], *λ* = $M_{w}h_{fg}$ is the molar latent heat of vaporization of water (J/mol) (*M_w_* is the molar mass of water kg/mol, *h_fg_* is the latent heat of evaporation (J/kg), *C* is the concentration of water inside the food (mol/m^3^), *ρ*_s_ is the density of the food product (kg/m^3^), m is the mass of moisture inside the food (kg), and *C_m_* is specific moisture capacity (kg_moisture_/kg_food_).

The value of *D_m_* is crucial to determine *Q_ev_*. *K_m_* is the moisture mass conductivity (kg·m^−1^·s^−1^). *Q_ev_* is calculated using the top-surface boundary condition in Equation (16).

The equation of mass conservation inside the food sample is shown in Equation (21), with the assumption of the negligible convection phenomena inside the food sample [24]:(21)$$\frac{\partial C}{\partial t}=D_{m}\nabla^{2}C$$
-Initial condition:
(22)$$\frac{\partial C}{\partial t}=0,\;\;\%moisture=76\%\;\left(wet\;basis\right),=>C_{0}=0.76\frac{\rho_{s}}{M_{w\;}},\;t=0\;\;\forall x,y,z,\;\;$$
-At the top surface of the sample, the boundary equation is expressed as:
(23)$$D_{m}.\frac{\partial C}{\partial n}=k_{c}\left(C_{b}-C\right),\;\;\forall x,y,\;\;\forall t>0$$
(24)$$C_{b}=\frac{\rho_{s}m_{e}}{M_{w}}$$
where *C_b_* is the equilibrium moisture concentration of the surrounding air (mol/m^3^), *m_e_* is the equilibrium moisture content of air (decimal wet basis), *C* is the moisture concentration on the surface of the food material (mol/m^3^), and *k_c_* is the mass transfer coefficient (m/s).

-At the bottom and the lateral surfaces, no evaporation occurred. The equation for this boundary is:


(25)
$$D_{m}\cdot \frac{\partial C}{\partial n}=0$$



-The moisture loss is calculated from the following expression:

(26)
$$m_{ml}=\frac{C_{0}\cdot V_{sample}-{∭}^{\;}C\;dv\cdot V_{sample}M_{w}}{m_{db}}\cdot 100$$



*k_c_* is an important parameter for the calculation in Equation (23). This parameter is calculated from the following relation [23]:(27)$$k_{c}=\frac{h_{m}}{\rho_{s}C_{m}}$$
where *h_m_* is the convective mass transfer coefficient (kg/(m^2^·s)) and *C_m_* is the specific moisture capacity (kg_moisture_/kg_food_).

*h_m_* is calculated using Lewis’s analogy between the mass- and heat-transfer coefficients [25,26].
(28)$$h_{m}=\frac{hc}{Cp_{a}}\;\cdot {\left[le\frac{h_{fg}}{h_{f}}\right]}^{1-0.575}$$

Equation (28) required some additional equations for the calculation, which are shown in Table 1.

### 3.5. Thermophysical and Dielectric Properties

The properties used in the electromagnetic simulation are shown in Table 2 and Table 3, assuming the dielectric properties of PET, PTFE, and Tylose^®^ are constant throughout the experiments.

The dielectric properties (***ε****’_r_* and ***ε****″_r_*) of pâté as a function of temperature are shown in Figure 4.

Some factors could explain the nature of the temperature-dependent dielectric properties of pâté, such as the denaturation of meat protein (myofibrillar, collagen, and other soluble proteins), and the presence of water, starch, fat, salt, and sugar [18,34,35,36]. There is an explanation for the increase in dielectric loss when the temperature increases, because of the presence of salt ions (Na^+^) inside the sample, which increases its electrical conductivity and dielectric loss [37]. The salt also influences the variation in dielectric loss as a function of temperature, which increases the dielectric losses along with the increase in temperature when the salt concentration is greater than 0.5% (for 915 MHz) [12].
foods-12-01187-t003_Table 3Table 3Electrical conductivities of metallic materials inside the waveguide.Surface BoundaryElectrical ConductivityReferenceAluminum3.774·10^7^ S/mCOMSOL^®^ databaseBrass1.59·10^7^ S/m[38]Copper5.99·10^7^ S/mCOMSOL^®^ database

The values of the thermal properties used in heat- and mass-transfer modeling are shown in Table 4 and Table 5.

Since the protein structure of pâté is denatured, depending on the sample temperature, the thermal conductivity of pâté changes as a function of temperature (Figure 5) [46].foods-12-01187-t005_Table 5Table 5Equations used to calculate the numerical parameters for the Lewis analogy.ParametersValue/UnitsReferences*RH*0.3[47]*% dry matter*23.52%Measure*me*0.02[22]*h*_0_2501·10^3^ J/kg[29]*Da*2.5·10^−8^ (m^2^·s^−1^)[48]*Km*1.29·10^−9^ (kg·m^−1^·s^−1^)[22]*Cp_d.a_*1005 J/(kg·K)[29]*Cp_m_*1870 J/(kg·K)[29]*C_m_*0.003 kg/kg[22,49]*Z*0.999[30]*α*1.0062[30]*β*3.14·10−8 P−1[30]*γ*5.6·10−7 K−2[30]*A*1.2811805·10−5 K−2
[30]*B*−1.9509874·10−2 K−1
[30]*C*34.04926034[30]*D*−6.3536311·103 K
[30]

The values of the parameters used for the implementation of mass transfer are shown in Table 5. Those values were obtained from the literature review.

## 4. Model Design

### 4.1. Computational Details

The microwave-heating model was numerically solved using COMSOL^®^ Multiphysics 6.0. Coupled-heat and -mass transfers were implemented in COMSOL^®^ and the radio frequency module was used to model the electromagnetic field propagation. The frequency used in the model was 915 MHz, which was the same frequency generated by the solid-state generator during the experiment.

A two-step approach was used to solve the model. The initial temperature was firstly used to initialize the dielectric properties’ values of the materials and to predict the electromagnetic field distribution within the waveguide and microwave cavity (frequency domain). The results from this stationary study were then used as the initial value for a frequency-transient study. In this second step, strongly coupled equations were solved (Maxwell’s equations, heat- and mass-transfer equations) as the thermophysical and dielectric properties of the food samples were considered to be temperature-dependent.

The model was solved with a Dell^®^ PrecisionTM Workstation computer equipped with 2 × Intel^®^ Xeon processors (8 cores), at 2.5 GHz, with 256 GB RAM, running on Windows^®^ 8 Professional, 64 bits. The computation time took around 120 h to finish the whole model.

### 4.2. Mesh Configuration

For the electromagnetic field propagation, the maximum size of the mesh elements suggested by Zhang et al. [50] was to be less than half of the free-space wavelength. However, the moisture transfer on the surface of the sample required a much smaller mesh size due to the high water-concentration gradients. The narrow elements on the surfaces inside the cavity also required a smaller mesh size. The numbers of mesh elements used in this study were 293,581 tetrahedra, 9400 pyramids, 62,808 prisms, and 56,400 hexahedra, which was 422,189 elements in total, and its average quality of skewness was 0.6954. A sensitivity study was conducted to obtain the mesh independence of the numerical results.

### 4.3. Validation Method

The predicted temperatures within the samples were validated by comparison to the experimental results by using the root-mean-square error (*RMSE*) from Pitchai et al. [51] and Siguemoto et al. [10].
(29)$$RMSE=\sqrt{\frac{1}{n}\overset{n}{\underset{i=1}{\sum }}{\left(X_{1}-X_{2}\right)}^{2}}$$
where *X*_1_ and *X*_2_ are the values for analyzing the parameters of the compared data.

The experimental and modeling results were analyzed with a chi-squared statistic test available in Microsoft Excel^®^; the *p*-values were compared with a significant confidence-level alpha value of 0.05.

## 5. Results and Discussion

### 5.1. Electromagnetic Validation

Since the local electric-field strength was difficult to measure during a microwave-heating process, the electromagnetic simulations were validated by comparing the measured values of the reflection coefficients, which were calculated from the reflected electric field at various SSC positions. The measured reflection coefficient for Tylose^®^ was compared with the results of the simulations, as shown in Figure 6. Although the similar reflection coefficient profile did not indicate the similarity of the electric-field distribution inside the sample, it confirmed a similar level of transmitted microwave power due to the interactions between the electric field and sample (dielectric properties) at each SSC position. Interestingly, both simulations and experiments showed the same SSC positions, which led to the minimum reflection coefficient (impedance-matching), and similar reflection coefficient values at each position. This result confirms the good agreement of the overall interaction between the electromagnetic field and sample for each SSC position. The simulation also showed a similar agreement while predicting the reflection coefficient of the pâté at various SSC positions, as shown in Figure 7. The overall *RMSE* of the prediction of the reflection coefficient was 0.05. From a statistical point of view, the numerical and experimental results were statistically indistinguishable (*p*-value > 0.05).

### 5.2. Temperature Validation

#### 5.2.1. Tylose

The model for the microwave heating of Tylose^®^ used the heat-transfer coefficient *hc* = 6 W/(m^2^·K), which was within the range derived from Kosky et al. [52]. Figure 8 shows the temperature profiles of three points (T1, T2, and T3) during the heating of Tylose^®^ at positions A and B. Since the duration of the ramp-up heating was short (only 387 s), the mass transfer was not included in the model prediction, which comprised only the coupling of electromagnetic waves and heat transfer. Only two sample positions were used to validate the robustness of the model for the temperature prediction, because the reflection coefficients for the C and D positions were high, which is not an ideal condition for the operation of microwave heating. Figure 8 shows the temperature distribution profiles of positions A and B, which are similar despite the different positions inside the cavity. Temperature T1 was significantly higher than T2 and T3 (*p*-value < 0.05), because T1 was located at the center of the sample where the electric-field distribution was greater compared to T2 and T3. Interestingly, the temperatures of T2 and T3 made no significant difference, despite a 0.3 cm-distance difference from their central positions. Statistically, the numerical and experimental results for the three points at both positions A and B were not significantly different (*p*-value > 0.05). Additionally, the overall *RMSE* of the temperature prediction at both positions (A and B) was 0.1621. However, the model appears to be slightly better at predicting the temperature profile at position B (*RMSE* = 0.1023) than at position A (*RMSE* = 0.2023). Overall, the numerical model exhibited a good agreement with the microwave heating of Tylose^®^.

Infrared thermography is a common technique that can also be used to validate the microwave-heating simulation of surface temperatures. Figure 9 shows the temperature cartography of the sample’s surface at position A during the microwave-heating process (27 W constant microwave-input power). At *t* = 0 min, the temperature of the PET near its wall was higher than 4 °C, due to the ambient natural convection with the surrounding air during the sample preparation. Although the temperature magnitude and sizes of the hot and cold spots were considerably different at the beginning of the experiment (*t* = 0 and *t* = 1 min), the experimental and numerical results were similar when the heating duration was increased. This result is supported further by a cut-line temperature-profile analysis, as shown in Figure 9.

The blue line in Figure 9 indicates the location of the temperature sampling line. The temperature profiles (IR images and simulation) were evaluated by comparing the temperature profiles along the line of the experimental data and simulations (Figure 10). The results show the reduction in the temperature difference between the experiments and simulations, while increasing the duration of the treatment. In these conditions, the effect of microwave heating becomes more dominant, reducing the small non-homogeneous initial temperature influence. This result confirms the negligible effect of water evaporation occurring at the surface for Tylose^®^ during the first 5 min of heating, since the model could provide an acceptable result without considering the mass-transfer phenomena.

After 5 min of microwave heating, the appearance of air bubbles due to water vapor at the surface of the Tylose^®^ deviated from the sample properties from the model’s assumptions. Thus, the microwave-heating experiment for Tylose^®^ was conducted for only 5 min for the validation of the model.

#### 5.2.2. Cambodian Pâté

Figure 11 shows the temperatures of T1, T2, and T3 during the microwave heating of the pâté at position A. Similar to the temperature profile of Tylose^®^ during heating, the temperature profile of T1 was significantly higher than T2 and T3 at the beginning, and both T2 and T3 had similar temperature profiles. The heating rate at each geometrical point was governed by the thermophysical properties of the sample (specific heat and thermal conductivity) and also the evolution of the electric-field strength at that point due to the temperature-dependence of the dielectric properties (Figure 4). Three convective heat-transfer coefficients on the upper surface were tested (*hc*_1_ = 3.5 W/(m^2^·K), *hc*_2_ = 6 W/(m^2^·K), and *hc*_3_ = 8.4 W/(m^2^·K)). For temperatures ranging from 4 to 80 °C, the heat-transfer coefficients were in the range of the natural convection predictions calculated from empirical correlations dedicated to free natural convections in the COMSOL program. During the ramp-up heating, the χ^2^ analyses of all *hc* values (3.5, 6, and 8.4 W/(m^2^.K)) exhibited insignificant differences, as compared with the experimental results (*p*-value > 0.05) with *RMSE* values of 3.5589, 2.3947, and 2.0000, respectively. This means that the simulation provided a better prediction of the temperature profile during the ramp-up heating for all heat-transfer coefficients that were in the studied range. Interestingly, a significant deviation between the temperature profiles of the simulation and experiment occurred during the temperature-holding phase, due to the effect of the evaporation flux that became noticeable, even at a low microwave-power input (around 11 W). The Lewis analogy calculation showed that the moisture-evaporation value calculated for *hc*_1_ was lower than the evaporation rate calculated for *hc*_3_, which resulted in reduced evaporative heat-loss values in the case of *hc*_1_, and it ultimately resulted in the temperature profile of *hc*_1_ being significantly higher than the temperature profiles of *hc*_2_ and *hc*_3_ during the temperature-holding phase (*RMSE* values of 15.8756, 12.6705, and 3.5048 compared to the experimental results, respectively). Therefore, the value of *hc*_3_ provided better temperature-prediction results during the temperature-holding phase. Overall, the *RMSE* between the experimental temperature results and predicted temperatures of *hc*_1_, *hc*_2_, and *hc*_3_, for both heating processes, were 11.3768, 9.0139, and 2.8357, respectively, which confirms that the *hc*_3_ value provides the best temperature prediction for both processes. During the temperature-holding phase, a small degree of deformation was observed experimentally due to water loss; however, the *RMSE* of the temperature prediction for *hc*_3_ showed that the simulation could still predict the temperature profile well, without taking into account the sample shrinkage in the model. Although the influence of the sample deformation was demonstrated in the study by Gulati, Zhu, and Datta [8], which showed its influence on the temperature profile during drying, the sample in the study lost a small amount of water that only led to a very small degree of deformation that did not have a significant interference with the model prediction. The *RMSE* analysis of the temperature prediction during the ramp-up heating phase of the pâté was higher than the *RMSE* for Tylose^®^; however, it still produced a good prediction regarding the duration of the microwave heating of the pâté, which was more than two-times longer.

According to the simulations (Figure 12), the maximum temperature difference between the hot and tiny cold-spot region was high (around 95 °C) following the ramp-up heating process (at *t* = 1880 s), which means that the pâté was not well-cooked when only using the ramp-up microwave-heating method. Nevertheless, this technique could be a good preheating method for the main cooking step because the duration of the ramp-up heating could be reduced by simply increasing the microwave power input. The maximum temperature difference was reduced to around 73 °C at the end of the temperature-holding phase (*t* = 3680 s). This improvement occurred due to the temperature-holding-phase duration with a modulated microwave power input that allowed the thermal conduction to reduce the large temperature gradient. The temperature was thus better distributed within the sample when being held at 80 °C for 30 min. This demonstrates that combining the ramp-up heating and temperature-holding phase is a better way to cook Cambodian pâté in terms of reducing the temperature gradients within the sample (Figure 12). Additionally, the cold spot is a very small region close to the top of the surface, which is submitted to both evaporation and natural convection with the surrounding air at the opposite face of the microwave incidence. The cold spot can be reduced by the rotation of the sample during heating or the addition of a metallic film resonator close to the surface [7,10].

Figure 13 shows the infrared images of the surface of the pâté during the experimental ramp-up heating process at 27 W of input power, and the results of the simulations. The shape and size of the hot and cold spots are shown in a similar 2D profile for the experiments and numerical results, which further confirms the validity of the model for the surface temperature. The shape of the hot spot also indicates that the temperature is unevenly distributed on the surface as microwaves are partially reflected from the sample due to its cylindrical shape and the variations in the dielectric properties inside the sample, which causes asymmetrical heating between both the side-facing antenna and SSC. The infrared images also confirm the non-uniformity of the temperatures during ramp-up heating, which is a concern for pâté cooking in terms of microbial inactivation and cooking homogeneity. It shows that constantly supplying microwave power is not the best way to cook the pâté.

Figure 14 shows the temperature profiles obtained from experiments and simulations. The results express similar trends. Similar to the Tylose^®^ cut-line temperature profiles, the initial temperature is fairly different; however, the temperature of the hot and cold spots evolved to be close to the temperature predicted by the simulation. The chi-squared analysis of both results are insignificantly different (*p*-value > 0.05).

### 5.3. Mass Transfer Validation

Since the top surface is the only surface exposed to the air surrounding the medium, it is the only surface from where the water vapor was evaporated. Figure 15 shows the increase in the percentage of moisture loss (% d.b.) during the combined heating process (ramp-up heating + temperature-holding phase). During 1880 s of ramp-up heating to 80 °C, the moisture loss was 16% d.b, and it increased to 28.5% d.b when the duration of the temperature-holding phase increased. Figure 15 (left) also shows the simulation results for the water concentration (% d.b) from the top surface to the bottom of the product along a central cut-line. The top surface had a low concentration of water because it was in a place where the water started to evaporate into the atmosphere, which drew other water molecules in the nearby region toward that surface due to the diffusion. The process continued to another nearby region, which resulted in a gradual reduction in the water concentration, as shown in Figure 16 and Figure 15. Moreover, the statistical analyses of the concentration profile of the center line received from *hc*_1_, *hc*_2_, and *hc*_3_ were significantly different (*p*-value < 0.05). Statistically, the percentage of global moisture loss due to the evaporation of water in the experiments and simulations were similar (*p*-value > 0.05) for all three *hc* values (*RMSE* values of 1.9219, 0.2862, and 0.5829, respectively). On the other hand, the reduced water concentration on the top surface of the pâté may be a concern in relation to the organoleptic quality of the product after cooking.

### 5.4. Sensitivity Analysis

The heat-transfer coefficient due to natural convection phenomena on the top surface can vary depending on the temperature variations on the surface during microwave cooking. Table 6 shows the *RMSE* values of simulated temperatures (T1, T2, and T3) obtained with various combinations of heat-transfer coefficients. First, Table 6 indicates the insignificant effect of heat-transfer coefficient variations on the temperature profiles during ramp-up heating. The statistical analysis of these temperature profiles is similar (*p*-value > 0.05) as the microwave absorbed the power that overcomes the effect of heat loss by convection and evaporation. However, the effect of both heat-transfer coefficient values becomes significant when the microwave power input is modulated during the temperature-holding phase, in which all three temperature profiles are significantly different (*p*-value < 0.05) with an *RMSE* value as shown in Table 6.

The comparison of the influence of *hc* without mass transfer shows that its variation does not significantly influence the prediction of the temperature profiles during the ramp-up heating process, as shown in Table 6.

When comparing the simulated temperatures of the model with and without mass transfer, Table 6 shows the significant influence of mass transfer in the predicted temperature profiles for both ramp-up heating and temperature-holding phases for the *hc*_3_ value. However, no significant effect was observed for the prediction of the temperature profile during ramp-up heating with a *hc*_1_ value for a similar model comparison. This was due to the low magnitude of *hc*_1_ leading to the low value of the mass transfer coefficient that was calculated using the Lewis analogy. However, at the temperature-holding phase, the temperature profile predicted by both models (with and without mass transfers) produced significant differences when calculated using *hc*_1_ and *hc*_3_ values.

## 6. Conclusions

The electromagnetic validation confirmed that the simulation exhibited good prediction results of the reflection coefficient profiles with an *RMSE* value of 0.050. The temperature prediction of Tylose^®^ during ramp-up heating was also in good agreement with the experiment (*RMSE* of 0.1621). Moreover, the result also shows that the temperature obtained from the simulation using an *hc*_3_ value is in agreement with the experimental result with an *RMSE* value of 2.0000 for the ramp-up heating and 3.5038 for the temperature-holding phase, while the overall prediction’s *RMSE* value for *hc*_3_ is 2.8357. The relevant validation of the developed model perfectly illustrated the use of a digital twin to improve the process. Indeed, it allowed us to show the following setbacks of microwave cooking. The 2D y–z-plan temperature profile during ramp-up heating meant that this cooking method did not entirely cook the sample because there was a large temperature gradient (95 °C) between the hot and cold spots; however, it can be used as a preheating method for further cooking processes. Moreover, the temperature-holding phase could improve the temperature uniformity inside the sample and reduce the temperature gradient to 73 °C. However, the temperature of a small part on the surface of the sample remained insufficiently low. Therefore, other improvements to temperature-uniformity methods, such as sample rotation and thin-film resonators, are also interesting for further exploration.

Based on the knowledge of the heat-transfer coefficient and microwave power input, the model was able to accurately predict the global moisture loss at the end of the ramp-up heating process (16.84% d.b) and temperature-holding phase (28.50% d.b). The analysis also quantified the significant deviations of the model predictions when mass-transfer phenomena are not taken into account during the microwave-cooking process. This comprehensive multiphysics model can be used, at present, as a digital twin to improve temperature homogeneity and reduce moisture loss during the 915 MHz microwave heating of Cambodian pâté. Moreover, the impact of water evaporation on the organoleptic properties of the product is also another interesting aspect for future studies to fully evaluate the microwave cooking of Cambodian pâté.

## Figures and Tables

**Figure 1 foods-12-01187-f001:**
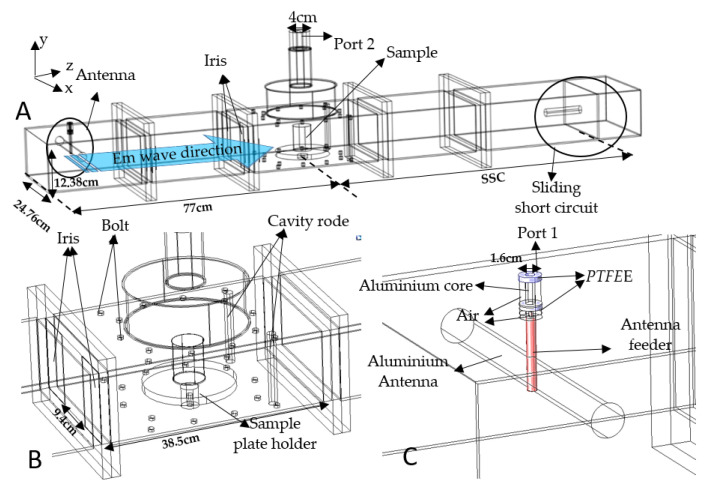
Detailed view of the microwave system’s (**A**) cavity (**B**) and antenna (**C**).

**Figure 2 foods-12-01187-f002:**
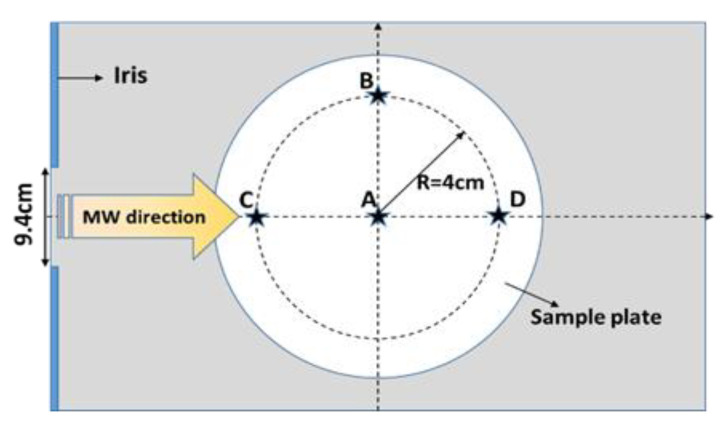
Sample positions within the single-mode microwave cavity (view from the top of the microwave cavity).

**Figure 3 foods-12-01187-f003:**
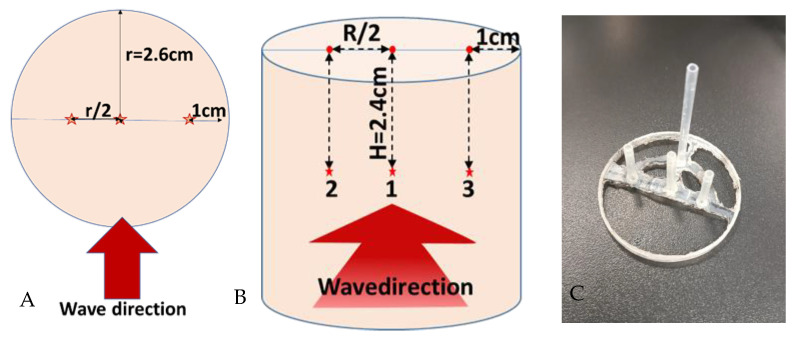
Position of optical fiber sensors in the sample (**A**,**B**) and sensor holder (**C**) (the red stars are the measured location, which is named 1, 2 and 3).

**Figure 4 foods-12-01187-f004:**
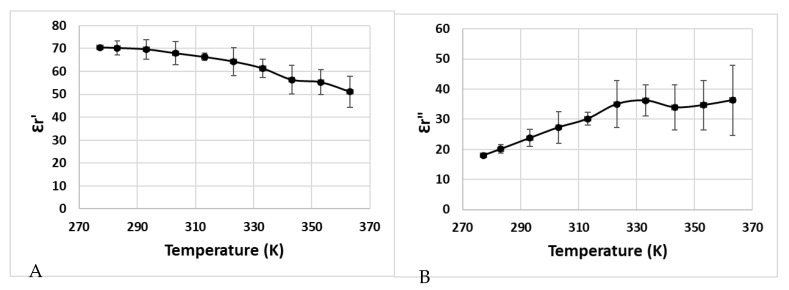
Dielectric properties of Cambodian pâté as a function of temperature at 915 MHz (**A**): measured value of Ԑr’(T), (**B**): measured value of Ԑr″(T)).

**Figure 5 foods-12-01187-f005:**
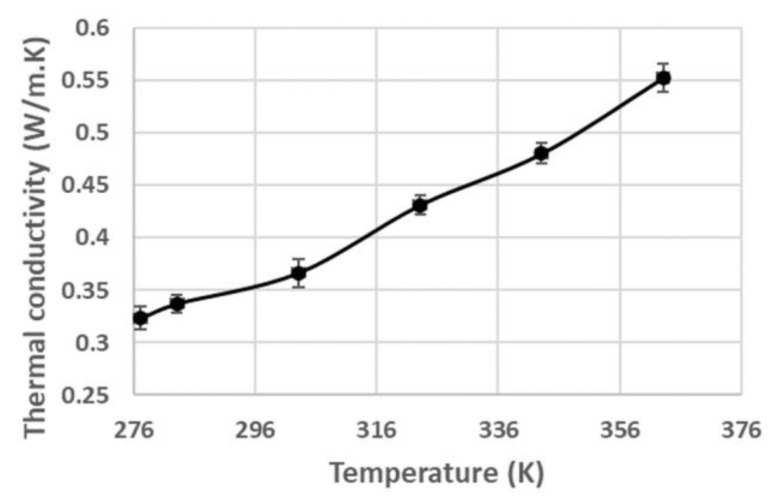
Thermal conductivity of pâté at various temperatures.

**Figure 6 foods-12-01187-f006:**
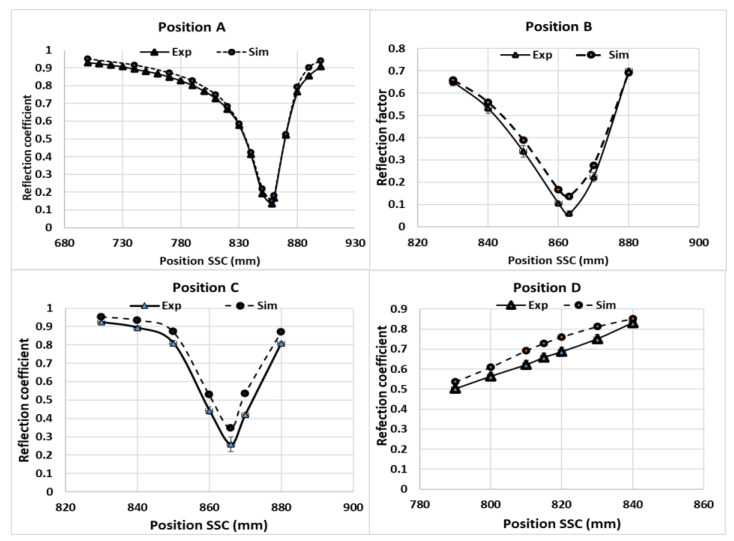
Reflection coefficients of Tylose^®^ at positions A, B, C, and D following various SSC positions.

**Figure 7 foods-12-01187-f007:**
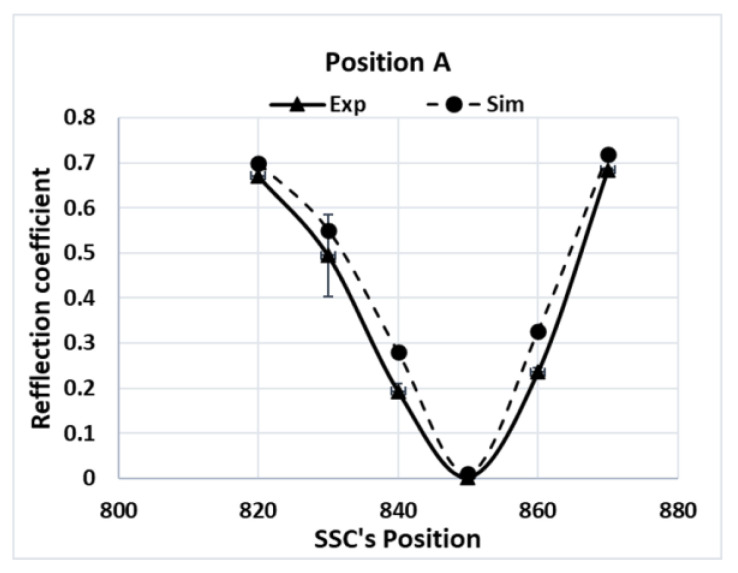
Reflection coefficients of pâté at position A for various SSC positions.

**Figure 8 foods-12-01187-f008:**
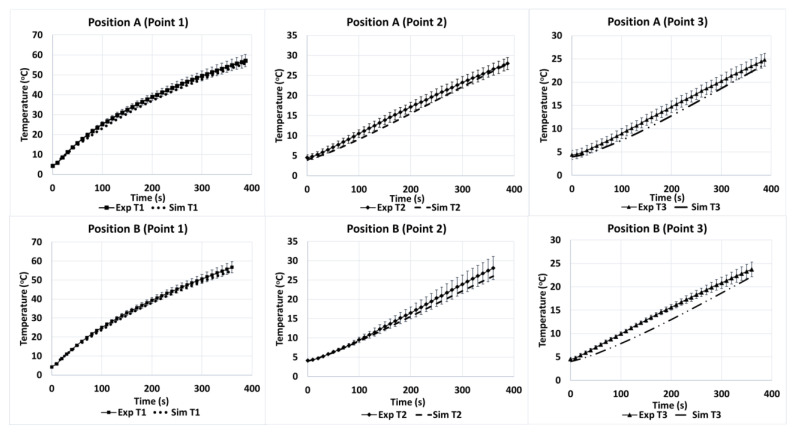
Temperature profiles of three points (1, 2 and 3) inside the Tylose^®^ sample during experimental heating at positions A and B (Exp) compared to simulation (Sim).

**Figure 9 foods-12-01187-f009:**
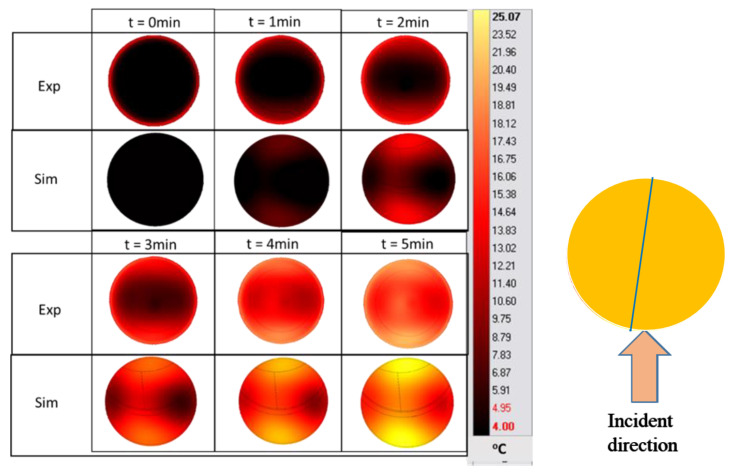
Surface-temperature cartography of Tylose^®^ during the ramp-up heating period at position A. (The blue line is the temperature sampling line used in Figure 10).

**Figure 10 foods-12-01187-f010:**
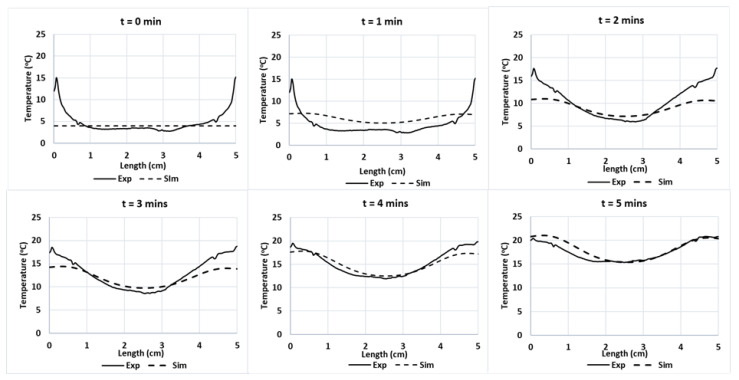
Experimental and predicted temperature profiles at Tylose^®^ sample’s surface along a cut-line during microwave heating during *t* = 0 to 5 min of microwave heating respectively.

**Figure 11 foods-12-01187-f011:**
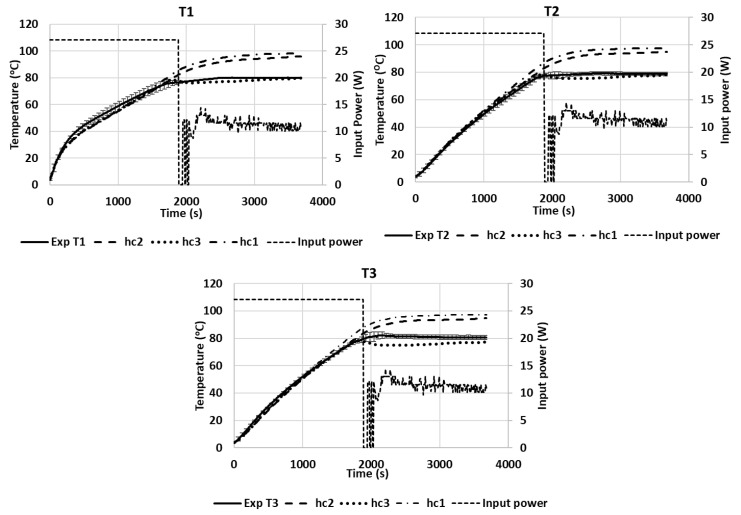
Temperature profiles of three points inside the pâté sample during microwave heating at positions A (Exp) compared with simulations.

**Figure 12 foods-12-01187-f012:**
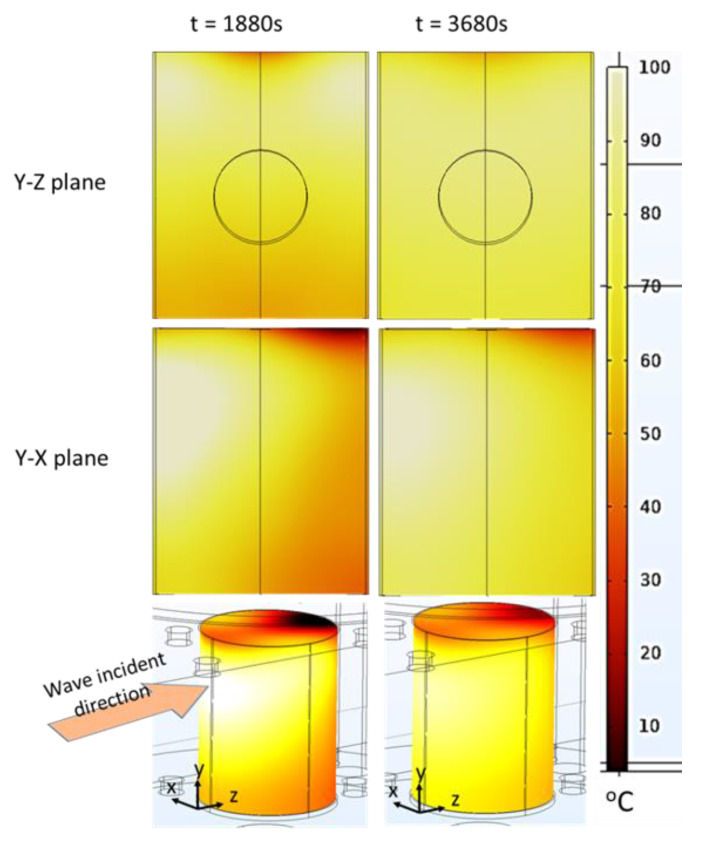
2D Temperature cartography at the end of the ramp-up heating (1880 s) process and temperature-holding phase (3680 s) (each plan passed through the centerline); 3D temperature map of the sample at 3680 s (simulation with *hc*_3_ value).

**Figure 13 foods-12-01187-f013:**
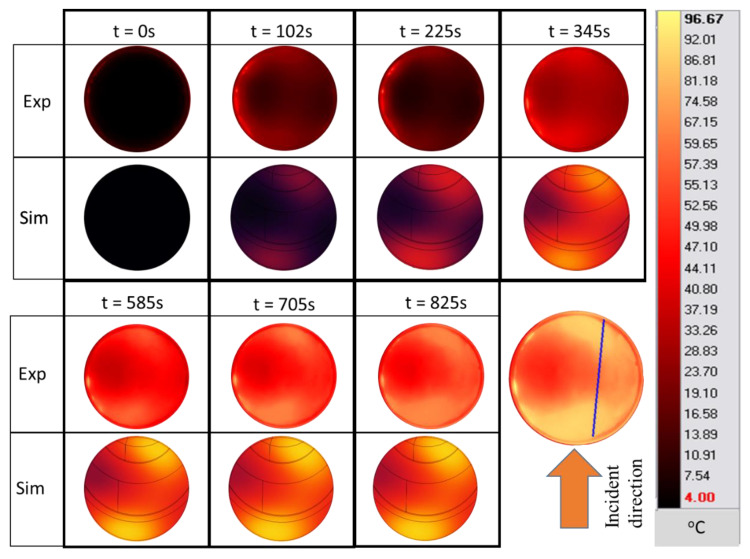
Surface temperature cartography of pâté during the ramp-up heating period at position A (simulation with *hc*_2_ value). (The blue line is the temperature sampling line used in Figure 14).

**Figure 14 foods-12-01187-f014:**
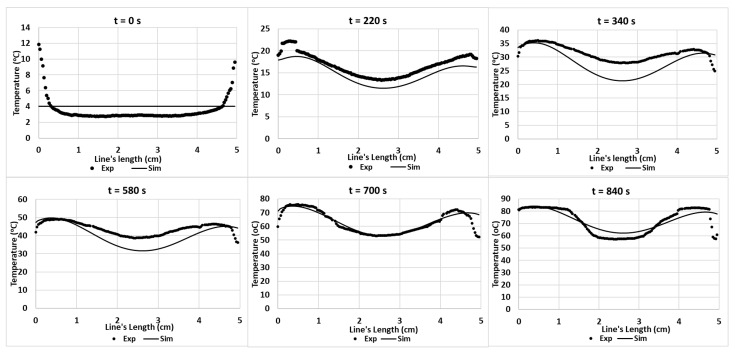
Experimental and predicted temperature profiles (using *hc*_3_ value) on the pâté’s surface along a cut-line during ramp-up heating.

**Figure 15 foods-12-01187-f015:**
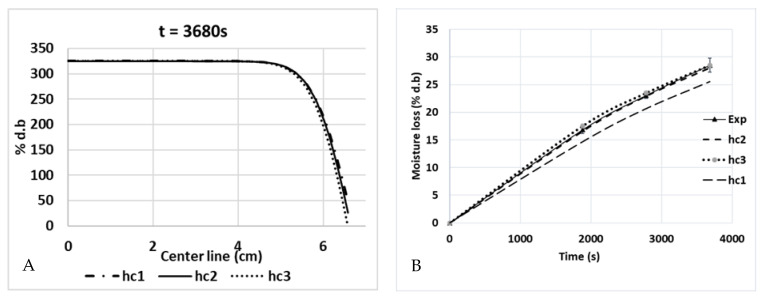
Water concentration profiles (% d.b.) along a central cut-line from the top surface to the bottom (**A**), and global moisture loss during cooking (**B**) (*hc*_1_ = 3.5 W/(m^2^·K), *hc*_2_ =6 W/(m^2^·K), and *hc*_3_ = 8.4 W/(m^2^·K)).

**Figure 16 foods-12-01187-f016:**
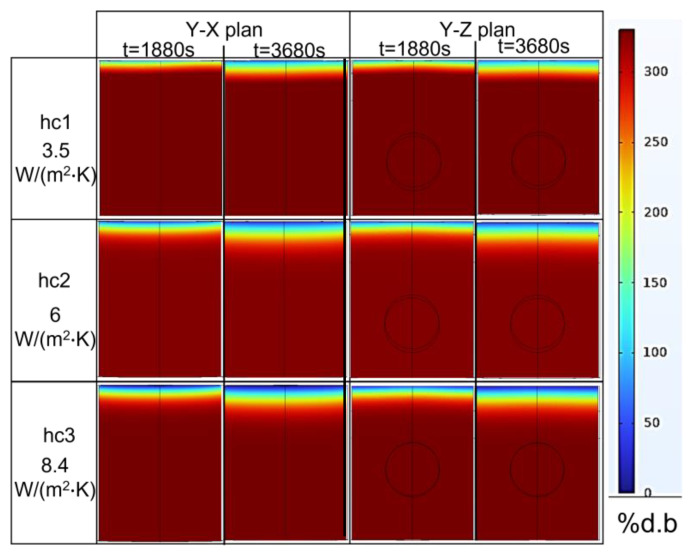
Two-dimensional plan of water concentration levels (% d.b.) at *t* = 1880 s and *t* = 3680 s for various heat-transfer coefficients on the surface of the pâté sample.

**Table 1 foods-12-01187-t001:** Equations used to calculate the numerical parameters for the Lewis analogy.

Name	Equation	Reference
Latent heat of vaporization of liquid water	$h_{fg}=1.91846\cdot 10^{6}{\left[\frac{T}{T-33.91}\right]}^{2}$	[27]
Lewis number	$le=\frac{k_{a}}{\rho_{a}Cp_{a}D_{a}}$	[28]
Heat capacity of the air around the sample	$Cp_{a}=Cp_{d.a}+d.Cp_{m}$	[28]
Humidity ratio in mass	$d=0.622\;\frac{RH\cdot p_{sv}}{p_{o}-RH\cdot p_{sv}}$	[29]
Thermal conductivity of air around the sample	$k_{a}=\left(251.626+7.734T_{a}+167.6x_{v}-7.432\cdot 10^{-4}T_{a}x_{v}+8.631\cdot 10^{-6}T_{a}{}^{2}\right)\cdot 10^{-5}$	[29]
The temperature of the hot-air film near the surface	$T_{a}=0.5\left(T_{air}+T_{s}\right)$	[30]
Mole fraction of moisture in the air	$x_{v}=RH\cdot f\prime \cdot p_{sv}\cdot P^{-1}$	[30]
Correction function	$f^{\prime }=\alpha +\beta p+\gamma {\left(T_{a}-273.15K\right)}^{2}$	[30]
Saturated pressure	$p_{sv}=1Pa\times \exp\left(AT_{a}{}^{2}+BT_{a}+C+DT_{a}{}^{-1}\right)$	[30]
Air density	$\rho_{a}=3.48353\cdot 10^{-3}\cdot \left(\frac{p}{ZT_{a}}\right)\left(1-0.378\;Xv\right)$	[30]
Enthalpy of hot-air film	$h_{f}=h_{a}+d\cdot h_{v}$	[29]
Enthalpy of dry air	$h_{a}=Cp_{a}\left(T_{a}-273.15K\right)$	[29]
Enthalpy of water vapor	$h_{v}=h_{0}+Cp_{v}\left(T_{a}-273.15K\right)$	[29]

**Table 2 foods-12-01187-t002:** Dielectric properties of materials at 915 MHz.

Material	*ε’r*	*ε″r*	Reference
PET container	3.5	0.0035	[31]
PTFE	2.1	0.002	[32]
Air	1	0	[33]
Tylose	69	8.9	Measurement

**Table 4 foods-12-01187-t004:** Thermophysical properties of the materials used for heat-transfer modeling.

Material	Density	Thermal Conductivity	Heat Capacity	References
Teflon	2.2·10^3^ kg/m^3^	0.32 W/(m.K)	1.02 J/(g.K)	[39,40]
Polypropylene	930 kg/m^3^	0.3 W/(m.K)	1.2 J/(g.K)	[41,42,43]
PET	1380 kg/m^3^	0.2 W/(m.K)	1.2 J/(g.K)	[44,45]
Tylose^®^	965 kg/m^3^	0.5 W/(m.K)	3859.9 J/(kg.K)	Measurement
Pâté	903.6 kg/m^3^	Figure 5	3511.3 J/(kg.K)	Measurement

**Table 6 foods-12-01187-t006:** *RMSE* and *p*-value of various comparisons of *hc* values of the model with and without mass transfer.

	Ramp-Up Heating *RMSE*/*p*-Value	Temperature-Holding Phase *RMSE*/*p*-Value
Including mass transfer		
*hc*_1_ and *hc*_3_	2.5603/*p*-value > 0.05	18.8437/*p*-value < 0.05
*hc*_2_ and *hc*_3_	0.9630/*p*-value > 0.05	15.5992/*p*-value < 0.05
*hc*_1_ and *hc*_2_	2.0257/*p*-value > 0.05	3.3303/*p*-value > 0.05
Without mass transfer		
*hc*_1_ and *hc*_3_	2.1473/*p*-value > 0.05	
Comparing models with and without mass transfer		
*hc* _1_	3.5680/*p*-value > 0.05	18.2840/*p*-value < 0.05
*hc* _3_	8.2854/*p*-value < 0.05	31.0090/*p*-value < 0.05
*hc*_1_ = 3.5 W/(m^2^·K), *hc*_2_ = 6 W/(m^2^·K), and *hc*_3_ = 8.4 W/(m^2^·K)).		

## Data Availability

The datasets generated for this study are available upon request from the corresponding author.

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
