# Peer review of "The Development of a Digital Twin to Improve the Quality and Safety Issues of Cambodian Pâté: The Application of 915 MHz Microwave Cooking"

_foods, 2023, doi:10.3390/foods12061187_

Round 1

Reviewer 1 Report

This paper is focused on the application of microwave heating/cooking of a model system simulating a traditional Cambodian pate, developing a model to predict temperature distribution and moisture loss of the model system and the pate during MW treatment. The results obtained by the Authors are very interesting for the process modeling and optimization.

I kindly ask the Authors to specify the role played  by the suggested process (MW) on the microbial count reduction of the investigated food, because They underlined in the Abstract, and also in the Introduction, the importance to guarantee the food safety. Furthermore, I ask the Authors if They have calculated the energy saving of the suggested treatment process in comparison with the traditional cooking method.

Please, control the unit of hc (line 410, 473)

Author Response

Dear reviewer 1

The attachment below is the response according to your comments.

Thank you for your comment.

Best regards

Sovannmony NGET

Reviewer 2 Report

The authors present interesting simulation results, with confirmation of experiments, on microwave heating of a 915 MHz single-modevity. The subject fits the theme of the journal. However, the manuscript is very long and requires a major improvement. Here are my specific comments/suggestions:

Introduction:

Line 39-63: There have been extensive computer simulation studies on microwave heating. What is your contribution? Instead of listing this literature on multi-mode cavities (domestic microwave ovens), I suggest the authors briefly introduce single-mode vs. multi-mode microwave cavities (their differences)? And why the authors designed and used a single mode cavity?

Line 62-74: the heating uniformity was not studied in this research, and no improvement method was explored. This literature review on microwave heating uniformity is irrelevant to this manuscript. Instead, I strongly suggest the authors consider solid-state vs. magnetron (a solid-state generator was used in this study). Solid-state microwave heating will be the future of food processing. In terms of computer simulation, it will be much easier and more accurate due to the stable and precise frequency spectrum of the solid-state generator. Please read https://doi.org/10.1016/j.ifset.2022.103240. A short literature review on magnetron-based microwave heating vs. solid-state heating (from a point of computer simulation) will be of more interest to readers. 

Line 75: "microwave heating of pate has never been studied before" is insufficient to state the significance of this study. What are the novelty and differences of your simulation work compared to previous studies? Please highlight.

Materials and Methods

Please use the past tense in writing.

Line 91: Please justify why you used a non-food to validate the simulation model.

Line 117-128: How did the authors calibrate the DP measurement device? What were measured frequency points (a single point or a frequency range)? Why were two different temperature ranges used for Tylose and pate samples?

Line 129: calibration of the probe?

Figure 1. 

1) Please indicate the E- and H-fields and wave propagation directions in Figure 1 (a);

2) The dimensions of the cavity, ports 1 and 2, and waveguides (length) should be given;

3) It is difficult to see Figures 1B & C due to too many unnecessary sketch lines. Suggest keeping the two drawings consistent with A (white background, same coordinate...)

Line 166-167: the use of the Iris needs to be explained in more detail. 

Line 182: Did the authors measure the S11 directly or measure reflected and incident power and then do the calculation? Please clearly state.

Line 187: the model, manufacturer, city, and country of the fiber optic sensors?

Line 200-203: what does it mean by "a microwave power regulation to hold the temperature of point at 80C"? Using microwave heating to hold sample temperature is impractical and uneconomical. Please justify.

Line 216-235: some of the assumptions may need support from literature.

Line 232: nature convection is not suited to solid food. Please remove it.

Line 268: Eq. (13) Please double-check whether there is a "2" before "Pi*f." Is E the time-average E-field intensity?

Line 354-367: Please indicate what microwave frequency used in the simulation (same as the set frequency of the solid-state generator?)

Line 366: what was the COMSOL computational time?

Results and discussions

Figure 6. Solid-state generators can adjust microwave frequency, making it straightforward for impedance matching. Why don't authors study the S11 as a function of frequency instead of SSC position? This would be more helpful for microwave coupling.

Line 432-433: How to measure infrared images should be clearly stated in Materials & Method.

Figure 9. What does the "blue line" mean (right side)? Also, what does it mean by the black lines on the sample surfaces (simulated temperature distribution)? Same thing is in Figure 13.

Figure 15, 16 and Table 6: Specify the convection coefficient values.

Overall, a major revision is needed.

Author Response

Dear reviewer 2,

The attachment file below is the response according to your comment.

Thank you for your comments.

Best regards,

Sovannmony NGET

Round 2

Reviewer 2 Report

The authors have addressed most of my concerns, and the manuscript has been improved. Here are comments/suggestions on the new content in the revised manuscript. 

Line 60: Reference [11], Zhou et al, instead of Zhu et al 

Line 128-129: Please specify the thermal conductivity of the calibration. Are the measured values in the range of the calibration?

Grammar issues: For example, 

Line 113 "...coaxial probe is performed..." should be "...coaxial probe was..."

Line 128: "the probe is first calibrated..." should be "the probe was ..."

These are just a few examples. The authors should correct all errors.

Author Response

Dear reviewer2,

Please kindly find the responses to your comment in the attachment file.

Thank you so much for your comment.

Best regards,

Sovannmony NGET
